Investigation of inhibition effect of daidzein on osteosarcoma cells based on experimental validation and systematic pharmacology analysis

Zhu Yufan
Yang Zhiqiang
Xie Yuanlong
Yang Min
Zhang Yufeng
Deng Zhouming dengzhouming@whu.edu.cn
Cai Lin orthopedics@whu.edu.cn
Department of Spine Surgery & Musculoskeletal Tumor, Zhongnan Hospital of Wuhan University , Wuhan , Hubei , People’s Republic of China
Costa-Lotufo Leticia
Electronic publication date: 2021 Aug 31
Publication date: 2021
Volume: 9
Electronic Location ID: e12072
Received 2021 Apr 7; Accepted 2021 Aug 5
Copyright: ©2021 Zhu et al.
Copyright year: 2021
Copyright holder: Zhu et al.
License: This is an open access article distributed under the terms of the Creative Commons Attribution License, which permits unrestricted use, distribution, reproduction and adaptation in any medium and for any purpose provided that it is properly attributed. For attribution, the original author(s), title, publication source (PeerJ) and either DOI or URL of the article must be cited.
License URL: https://creativecommons.org/licenses/by/4.0/

Keywords: Daidzein, Osteosarcoma, Systemic pharmacology, Src, ERK

Funding: The authors received no funding for this work.

==============================
Objective

This study aims to explore the effect of daidzein, which is a natural isoflavone compound mainly extracted from soybeans, on osteosarcoma and the potential molecular mechanism.

Material and Methods

143B and U2OS osteosarcoma cells were treated with gradient concentrations of daidzein, and MTT assay was used to determine the cell proliferation capacity and IC50. Hoechst 33342 staining and Annexin V-FITC/PI detection were used to determine apoptosis. Cell cycle was analyzed by flow cytometry, and migration ability were detected by transwell assays and scratch wound assay. An osteosarcoma xenograft mice model was applied to investigate the effect of daidzein on osteosarcoma in vivo. Systematic pharmacology and molecular modeling analysis were applied to predict the target of daidzein to osteosarcoma, and the target Src was verified by western blotting. We also observed the effect of daidzein on cell proliferation and apoptosis of Src-overexpressing osteosarcoma cells.

Results

In vitro, daidzein significantly inhibited 143B and U2OS osteosarcoma cell proliferation and migration, and induced cell cycle arrest. In vivo, daidzein exerts antitumor effects in osteosarcoma xenograft mice. After systematic screening and analysis, Src-MAPK signaling pathway was predicted as the highest-ranked pathway. Western blot demonstrated that daidzein inhibited phosphorylation of the Src-ERK pathway in osteosarcoma cells. Also, overexpression of Src could partially reverse the inhibitory effects of daidzein on osteosarcoma cell proliferation.

Conclusion

Daidzein exerts an antitumor effect on osteosarcoma, and the mechanism may be through the Src-ERK pathway.

Introduction

Osteosarcoma is the most common primary tumor of bone in children and adolescent. The 5-year survival rate of patients with localized osteosarcoma is approximately 70%, whereas the overall survival rate of patients with metastatic or recurrent disease is less than 20% (Harrison et al., 2018). Based on historical data, over 80% of patients will develop metastasis following resection of the primary tumor alone, and even with the addition of chemotherapy to primary tumor resection, approximately one-third of patients presenting with localized disease will subsequently develop pulmonary metastases (Khanna et al., 2014). Standard treatment modalities consist of combining surgical excision with chemotherapy, which has changed very little over the past 30 years (Gianferante, Mirabello & Savage, 2017; Li et al., 2020; Luu & Viloria-Petit, 2020). The three-drug chemotherapy regimen of cisplatin, doxorubicin, and methotrexate (MAP) forms the backbone therapy, and the response to induction chemotherapy correlates with patient outcome (Kleinerman, 2016). Due to the clinical heterogeneity of osteosarcoma, the selection of treatment options and effective drug combinations remain limited. It is generally accepted that plants are regarded as valuable resources for the development of drugs to treat many diseases, including cancer (Thomford et al., 2018). Some natural active ingredients display strong antitumor activity; in addition, compared with chemotherapy, they have fewer adverse effects and no resistance. Therefore, there is an urgent requirement to examine novel and more effective natural compounds for the treatment of osteosarcoma.

Flavonoids are phenolic compounds with more than 6,00 family members that widely exist in a large variety of plants and fungi. Isoflavone is a natural flavonoid that plays the role of phytoestrogens in mammals and has become a research hotspot in recent years (Zhang et al., 2018). Daidzein, whose chemical composition is 7,4-dihydroxyisoflavone, belongs to the isoflavone family. It is the most frequently ingested and deeply studied phytoestrogen (Jin et al., 2010). In previous studies, daidzein has been proven to have antitumor effects on a variety of tumors, including breast cancer (Bao et al., 2014; Magee et al., 2014), bladder cancer (He et al., 2016), prostate cancer (Singh-Gupta et al., 2010; Yu, Blackburn & Zhou, 2003), colon cancer (Liang et al., 2018), endometrial carcinogenesis (Lian et al., 2001), melanoma (Wang et al., 2002), neuroblastoma (Lo, Mak & Leung, 2007), and choriocarcinoma (Zheng et al., 2017). Recently, Luisa Salvatori et al. reported that daidzein inhibited proliferation and cell cycle progression and promoted apoptosis in a U2OS human osteosarcoma cell line in vitro (Salvatori et al., 2009), but there has been no research on the related mechanism. In addition, no other study has demonstrated that daidzein is capable of inhibiting osteosarcoma. Therefore, we carried out an in-depth study on the effect and mechanisms of daidzein against osteosarcoma.

Systemic pharmacology, proposed by Andrew L. Hopkins, is an emerging discipline that integrates systems biology and pharmacology (Hopkins, 2008). It provides a new way to explore the mechanism of drugs from the molecular and cellular levels to the tissue and organism levels (Berger & Iyengar, 2009). After verifying the inhibitory effects of daidzein on osteosarcoma in vitro and in vivo, we applied systemic pharmacology theory and molecular docking analysis to identify the intersection of potential targets between osteosarcoma and daidzein, and selected the key target, Src, and its downstream MAPK pathway. Finally, a rescue experiment was carried out to validate the mechanisms of action. The methodological flowchart of this study is shown in Fig. 1.

Figure 1 The methodological flowchart of this study.

This study firstly experimentally demonstrated the therapeutic effect of daidzein on osteosarcoma, then through the exploration of network pharmacology and molecular docking technology, key molecules and pathways were targeted and experimentally validated. Website links: DigSeE (Kim et al., 2013), DisGeNET (http://www.disgenet.org/), GeneCards (https://www.genecards.org/) KEGG disease (https://www.kegg.jp/kegg/disease/) MalaCards (http://www.malacards.org/) Phenopedia (https://phgkb.cdc.gov/PHGKB/startPagePhenoPedia.action) PubChem (https://pubchem.ncbi.nlm.nih.gov/) PharmMapper (http://www.lilab-ecust.cn/pharmmapper/) VENNY 2.1.0 (https://bioinfogp.cnb.csic.es/tools/venny/).

Materials & Methods

Reagents

Daidzein [7-hydroxy-3- (4-hydroxyphenyl) chromen-4-one] was purchased from Aladdin Industrial, Inc. (Shanghai, China), which with a purity over 98%. It was dissolved in DMSO at a final concentration of 100 mM in medium and stored at −20 °C. Dimethyl sulfoxide (DMSO) was obtained from Sinopharm Chemical Reagent Co., Ltd. (China).

Cell culture

Human osteosarcoma cell lines (143B and U2OS) and human osteoblast cell line hFOB 1.19 were obtained from the China Centre for Type Culture Collection (Wuhan, China). 143B cells were cultured in Roswell Park Memorial Institute (RPMI) 1640 medium supplemented with 10% heat - inactivated FBS (Gibco) and 1% antibiotics in a 37 °C incubator with 5% CO2. U2OS cells were cultured in McCoy’s 5A medium (modified) when hFOB 1.19 was cultured in α-MEM (HyClone) and the other conditions were the same as those used for 143B cells.

Cell proliferation analysis

Cells were plated in 96-well plates at a density of 2000 cells per well and incubated at 37 °C for 12 h. Then, the cells were treated with different concentrations of daidzein diluted in medium containing 10% FBS. After incubation, the supernatants were removed, and MTT solution (Beyotime, Shanghai, China) was added to each well (100 µL per well). Following incubation at 37 °C in a dark environment for an additional 4 h, the supernatants were removed again, and 200 µL DMSO was added to each well to solubilize formazan. After 10 min, the 96-well plates were placed in a microplate reader, and the absorbance was recorded at 570 nm. The data came from three independent experiments, each with six copies. The mean cell viability of the treated cells was used to calculate the survival rate.

Colony formation assay

The cells were evenly seeded in a 6-well plate at a density of 500 cells per well. After 24 h, the cells were treated with different concentrations of daidzein for approximately 7 days until the cells grew into visible colonies. Then, the supernatant was discarded, and the cells were washed with PBS three times. Colonies were then fixed with 4% paraformaldehyde for 30 min and stained with 1% crystal violet for 10 min. The number of colonies (more than 50 cells/colony) was counted and quantified.

Cell apoptosis and cycle detection

Following exposure to daidzein for a specified period, osteosarcoma cells were collected for apoptosis and cycle detection. Hoechst 33342 (Beyotime, Shanghai, China) was used to detect cell apoptosis by observing the cell morphology. After daidzein treatment, the medium was discarded, and the cells were washed twice with PBS and then fixed with sufficient 4% paraformaldehyde at 4 °C for 10 min. Then, sufficient Hoechst 33342 staining solution was added to cover the cells. After incubation in the dark for 10 min, the cells were washed with PBS and then immediately observed and imaged by fluorescence microscopy. In addition, cell apoptosis and cell cycle progression were analyzed by flow cytometry. Cell apoptosis of human osteosarcoma cells was analyzed based on Annexin V-FITC/propidium iodide (PI) dual staining when the cell cycle was assessed by staining cells with 20 µg/mL PI.

Migration ability detection

The migration ability of osteosarcoma cells was detected by Transwell migration assay and scratch wound assay. The Transwell experimental procedure was as follows: Approximately 5 ×104 cells suspended in 100 µL serum-free RPMI 1640 or McCoy’s 5A medium were added to each upper chamber, and 600 µL of medium containing 10% FBS was added to the lower chamber. Cells were cultured at 37 °C for 24 h, fixed with 4% paraformaldehyde for 30 min and stained with 1% crystal violet for 10 min. Cotton swabs were used to remove the unmigrated cells on the upper surface of the membrane, and the cells that migrated on the lower surface of the film were counted. As for scratch wound assays, cells were plated in a 6-well plate at a density of 5 ×105 cells per well and then incubated at 37 °C until almost complete confluence. Three parallel wounds were made with a 10 µL pipette tip per well. Cells were cultured in serum-free medium supplemented with different concentrations of daidzein. At 0 h and 24 h, photos were taken at the same position under an optical microscope, and the wound width of each group was compared.

Establishment of a nude mouse osteosarcoma xenograft model

Twelve four-week-old male BALB/c nude mice obtained from Hubei Experimental Animal Research Center (Hubei, China) were used for animal experiments. 143B cells were digested and suspended in cold PBS at a final concentration of 1 ×107 cells per mL. In this model, a total of 100 µL of the prepared cell suspension was injected in the right subaxillary area subcutaneously. 7 days after tumor inoculation, when the volume of the tumor was approximately 100 mm3, daidzein (20 mg/kg) was administered by tail vein injection every 2 days in the experimental group (n = 6), while the control group (n = 6) was injected with an equal volume of DMSO. The injection treatment lasted for 9 days, and the tumor volume and mouse body weight were measured every 3 days. Tumor volume (mm3) was calculated using the following equation: length × width2 × 0.5. On day 16, mice were sacrificed by dislocation of cervical vertebra after anesthesia with 1% sodium pentobarbital (100 mg/kg body weight, intraperitoneal injection), and the xenografts were dissected and fixed with 4% polyformaldehyde for H&E staining analysis. When the tumor diameter exceeded 20 mm or rupture and infection occurred, as well as when treatment was ineffective after systemic infection occurred, mice were euthanized using the method described above.

This in vivo study was conducted according to the guidelines of the Declaration of Helsinki, and approved by the Institutional Review Board (or Ethics Committee) of Institutional Animal Care and Use Committee, Wuhan University Center for Animal Experiment (protocol code WP2020-08106. On August 10th, 2020).

Target identification and network construction

The osteosarcoma-related targets were collected and combined from the following databases: DigSeE (Kim et al., 2013), DisGeNET (http://www.disgenet.org/), GeneCards (https://www.genecards.org/), KEGG disease (https://www.kegg.jp/kegg/disease/), MalaCards (http://www.malacards.org/), and Phenopedia (https://phgkb.cdc.gov/PHGKB/startPagePhenoPedia.action). The validated targets of daidzein were extracted from PubChem (https://pubchem.ncbi.nlm.nih.gov/) and PharmMapper (http://www.lilab-ecust.cn/pharmmapper/). Common targets of osteosarcoma and daidzein were discovered and retained by VENNY 2.1.0 (https://bioinfogp.cnb.csic.es/tools/venny/) for further analysis. A protein-protein interaction (PPI) network was constructed by STRING (https://string-db.org/) and visualized by Cytoscape software. The Database for Annotation, Visualization and Integrated Discovery (DAVID; http://david.abcc.ncifcrf.gov/) was used to perform GO and pathway enrichment analyses, and Cytoscape software (Version 3.8.1) was used to construct the target-function network.

Molecular docking studies

To identify the potential interaction between Src and daidzein, molecular docking simulations were conducted with AutoDock Vina (version 1.1.2) to predict their preferred binding conformation. PubChem was referenced for the SDF file of daidzein, and the SDF file was transformed into a PDB file by Open Babel 2.3.2. The crystal structure of human Src was derived from the Protein Data Bank (PDB) database (https://www.rcsb.org/structure/2H8H). Before molecular docking analysis, PYMOL 2.3.4 software was used to remove the water molecules as well as ligands from the receptor protein, and AutodockTools software was used to balance the charge and affix hydrogen atoms to the receptor protein. Grid Option was used to adjust the number of grid points in each direction, the center of the binding pocket and the spacing between the grid points. The centroid of the daidzein molecule in the crystal structures of the complex was defined as the binding site. AutoDock Vina software was used for molecular docking of the Src protein and daidzein molecule, and the results were output in the form of affinity. The affinity is obtained by calculating the space effect, repulsion, hydrogen bond, hydrophobic interaction and molecular flexibility of the complex, which is an important indicator to measure whether the ligand can effectively bind to the receptor molecule. The lower the value, the better the binding effect.

Analysis of proteins in osteosarcoma cells treated with daidzein

Cold PBS was used to wash the adherent osteosarcoma cells cultured in 6-well plates three times. RIPA buffer containing a proteinase inhibitor (Beyotime, Shanghai, China) was used to extract total protein from osteosarcoma cells. Osteosarcoma cells were scraped with a cell scraper, dissolved in buffer, and transferred to a 1.5 mL EP tube with a pipette. The above samples were treated with an ultrasonic cell breaker, with ultrasonic treatment for 3s and an interval time of 10s, which was repeated three times. Then, the lysates were centrifuged at 4 °C and 13000 g for 15 min. In this way, the protein composition was separated and diluted for denaturation. Protein samples were subjected to SDS-PAGE and then transferred to PVDF membranes. The blocking solution was prepared by dissolving 5% skim milk powder in TBST, and the membrane was put into the solution and blocked for 1 h. The membranes were then incubated with the following rabbit anti-human polyclonal primary antibodies overnight: anti-Src (ab109381), anti-p-Src (ab40660, phospho Y418), anti-JNK (ab179461), anti-p-JNK (ab76572, phospho Y185+Y185+Y223), anti-ERK (ab184699), anti-p-ERK (ab201015, phospho T202+T185), anti-p38 (ab178867, phospho T180), anti-p-p38 (ab195049, phospho T180+Y182) and anti-GAPDH (ab9485) (Abcam, Shanghai, China). GAPDH expression was used as a loading control. After washing three times, the membranes were incubated with the corresponding goat anti-rabbit secondary antibodies (Abcam, Shanghai, China). A Western blot visualization (ECL kit, Pierce, USA) and analysis system (Bio-Rad) was used to detect chemiluminescence.

Construction of Src-overexpressing osteosarcoma cell lines by lentivirus transfection

For Src overexpression, 143B and U2OS cells were transfected with 4 µg of pMSCV-Flag or pMSCV-Flag-Src using Lipofectamine 3000 Transfection Reagent (Thermo Scientific, USA) according to the manufacturer’s instructions. Empty virus vectors were used as negative controls. The primers were 5′-tgttcggaggcttcaactcc-3′(F) and 5′-gacatccaccttcctcgtgt-3′(R). After 72 h of transfection, the expression of the green fluorescent protein EGFP in cells was observed under a fluorescence microscope, and stably transfected cell lines were established by the puromycin (1.5 ng/µL) screening method. The rescue experiment was performed based on stably transfected cells.

Statistical analysis

The results are expressed as the mean ± standard deviation (SD) from at least three independent experiments. Student’s t-test was used to compared the difference between two groups while, one-way analysis of variance (ANOVA) was used to analyze statistical differences between more than two groups. GraphPad Prism software (version 7.04) was used for statistical analysis. A statistically significant difference was defined as p < 0.05.

Results

Daidzein inhibits the proliferation of 143B and U2OS osteosarcoma cells

Firstly, we explored the effect of daidzein on proliferation of 143B and U2OS osteosarcoma cells. 143B and U2OS cells were treated with gradient concentrations of daidzein (0, 10, 20, 50, 100, 200 or 500 µmol/L). After 48 or 72 h treatment, MTT assay was performed to detect the cell proliferation capacity. As shown in Figs. 2A–2B, daidzein significantly inhibited the proliferation capacity of osteosarcoma cells in a dose- and time-dependent manner. According to the absorbance curve, the 48 h 50% inhibiting concentration (IC50) of daidzein on 143B and U2OS cells are 63.59 µmol/L and 125 µmol/L. Meanwhile, we treated normal human hFOB 1.19 osteoblasts with the same daidzein dose in 48 and 72 h (Fig. 2C). According to the analysis results of IC50 (Fig. 2D), daidzein had a much weaker inhibitory effect on the proliferation of hFOB cells compared with osteosarcoma cells (Student’s t-test, p < 0.05). Thus, daidzein should be considered relatively safe when applied clinically. Then, we set three concentration treatment groups (143B: 20 µmol/L for the low concentration group, 48 h IC50 group, and 200 µmol/L for the high concentration group; U2OS: 50 µmol/L for the low concentration group, 48 h IC50 group, and 200 µmol/L for the high concentration group) for the following experiments. And colony formation assay results also showed that daidzein significantly inhibited the colony formation ability of 143B and U2OS osteosarcoma cells (Figs. 2E–2F). Thus, daidzein may inhibits the proliferation of 143B and U2OS osteosarcoma cells in a dose- and time-dependent manner.

Figure 2 Daidzein inhibited cell proliferation and induced apoptosis in vitro.

(A) MTT analysis of 143B and U2OS cells treated with different concentrations of daidzein for 48 or 72 h. MTT assay was performed in sextuplicate. Dose- and time-dependent inhibition of cell growth could be observed (P < 0.0001, ANOVA analysis). (B) IC50 values in 143B and U2OS cells after 48 or 72 h of daidzein treatment. (C) MTT analysis of hFOB cells treated with different concentrations of daidzein for 48 or 72 h. (D) IC50 values in non-tumor cells hFOB after 48 or 72 h of daidzein treatment. (E) Seven-day colony formation assay of 143B and U2OS cells treated with different concentrations of daidzein. (F) Colony counts in 143B and U2OS cells treated with different concentrations of daidzein from four independent experiments. (G) Hoechst 33342 staining assay of 143B and U2OS cells treated with different concentrations of daidzein. (Solid bars and errors represent the mean ± SEM. Individual values were shown as dots of different shapes. Student’s t-test, ∗p < 0.05, ∗∗p < 0.01, ∗∗∗p < 0.001, ∗∗∗∗p < 0.0001 versus control group).

Daidzein induces apoptosis and cell cycle arrest in 143B and U2OS osteosarcoma cells

Hoechst 33342 staining and Annexin V-FITC/PI dual staining detection kits were used to determine the effect of daidzein on apoptosis 143B and U2OS cells. The results indicated that 24 h of treatment with daidzein (at low, IC50, and high concentrations) considerably increased apoptosis in a dose-dependent manner compared to that of the control group. The results of the flow cytometric analysis showed that the early and late apoptosis rate of daidzein treatment group (at low, IC50, and high concentrations) were significantly lower than that of the control group (Figs. 3A–3B). Next, we determined whether daidzein was involved in the cell cycle of osteosarcoma cells. According to the cell cycle distribution analyzed by flow cytometry, we found that daidzein increased the percentage of S phase cells but decreased the percentage of G0/G1 phase cells in 143B and U2OS cells (Figs. 3C–3D). These results suggest that daidzein may induce apoptosis and cell cycle arrest in 143B and U2OS osteosarcoma cells.

Daidzein suppresses migration ability of and 143B and U2OS osteosarcoma cells

We also explored the influence of daidzein on migration ability of osteosarcoma. 143B and U2OS cells were pretreated with daidzein for 24 h before the Transwell assays. Daidzein significantly suppressed cell migration in 143B and U2OS cells, as shown in Fig. 3E. Moreover, scratch wound assay revealed that the cell mobility rates in daidzein treatment group were significantly decreased when compared with the control group (Fig. 3G). Therefore, daidzein can suppress migration ability of and 143B and U2OS osteosarcoma cells.

Figure 3 Incubation of osteosarcoma cells with daidzein promoted apoptosis and cell cycle arrest when inhibited migration.

(A) 143B and U2OS cells treated with different concentrations of daidzein were stained with Annexin V-FITC/PI, analyzed by flow cytometry. Quantitative analysis of apoptotic cells from three independent experiments. (B) Cell cycle detection of 143B and U2OS cells treated with different concentrations of daidzein, examined by flow cytometry. (C) Quantitative analysis of cycle distribution from three independent experiments. (D) Transwell assay of 143B and U2OS cells pretreated with different concentrations of daidzein. (F) Invaded cell counts in 143B and U2OS cells pretreated with different concentrations of daidzein. (E) Scratch wound migration analysis of osteosarcoma cells treated with different concentrations of daidzein. (F) Wound closure rates in osteosarcoma cells treated with different concentrations of daidzein. (Solid bars and errors represent the mean ± SEM, Individual values were shown as dots of different shapes, Student’s t-test, ∗p < 0.05, ∗∗p < 0.01, ∗∗∗p < 0.001, ∗∗∗∗p < 0.0001 versus control group).

Daidzein effectively suppresses osteosarcoma growth in xenograft mice model

To investigate the effect of daidzein on osteosarcoma in vivo, an osteosarcoma xenograft mice model was established by subcutaneous implantation of 143B cells in BALB/c nude mice. As shown in Fig. 4, the volume and weight of the tumors and tumor growth rate in the daidzein-treated group significantly decreased when compared with those in the control group. H&E staining of the tumor revealed that the number of necrotic cells in the daidzein-treated group was significantly higher than that in the control group (Fig. 4E). Furthermore, the body weight of the nude mice was not significantly affected (Fig. 4B), indicating that there was little systemic toxicity of daidzein in vivo.

Figure 4 Daidzein exerts antitumor effects in vivo without obvious toxicity.

(A) Gross observation of osteosarcoma xenograft mice on the day the mice were sacrificed. (B) Body weight during the daidzein injection. There was no significant difference between the two groups. (ANOVA analysis) (C) Gross observation of tumor tissue on the day the mice were sacrificed. (D) Tumor volume during the daidzein injection. Time-dependent inhibition of cell growth could be observed (P < 0.0001, ANOVA analysis) (E) H&E staining of the tumor. The daidzein group showed obvious tumor necrosis. (F) Tumor weight on day 16. Solid bars and errors represent the mean ± SEM, Student’s t-test, ∗∗∗∗p < 0.0001 versus control group.

Target prediction of daidzein on osteosarcoma through systemic pharmacology analysis

To reveal the mechanism of daidzein regulating biological process of osteosarcoma cells, systemic pharmacology analysis was performed on PharmMapper platform. A total of 243 human genes associated with daidzein were predicted by the PharmMapper platform (Table S1). As shown in Fig. 5A, a “daidzein-daidzein targets” network was constructed. Osteosarcoma-related genes were collected and combined from DigSeE, DisGeNET, GeneCards, KEGG disease, MalaCards, and Phenopedia (Table S2). After intersecting the two gene sets (Fig. 5B), there were 133 common targets of osteosarcoma and daidzein through which daidzein might play a role in regulating the progression of osteosarcoma. After that, 133 target genes associated with daidzein and osteosarcoma were imported into the STRING database for PPI network construction. As shown in Fig. 5C and Table S3, there were 133 nodes and 295 interaction edges (interaction score>0.9). To analyze the PPI diagram, the analysis tool in Cytoscape was applied, and the top 30 hub genes were identified and are shown in Fig. 5D. Among the top 30 hub genes, Src was the node that interacted with the most numerous edges, suggesting that Src may play a vital role of daidzein in regulating biological process of osteosarcoma.

Figure 5 Src is the most likely target molecule for direct binding by daidzein.

(A) Network of daidzein-target genes. Purple diamonds represent daidzein; green circles represent target genes of daidzein. (B) Venn diagram revealing the common target genes of daidzein against osteosarcoma. (C) Protein-protein interaction (PPI) network. Nodes: common targets of daidzein and osteosarcoma; Edges: PPIs between targets of daidzein and their interaction partners. (D) The top 30 hub genes in the PPI network that interacted with the most numerous edges. (E) The Ball and Stick 3D structure of daidzein. The gray ball represents the carbon atom when the red ball represents the oxygen atom, and the small white ball represents the hydrogen atom. (F) Molecular docking simulation predicted the amino acid residues bound by daidzein. (G) Molecular docking simulation of daidzein to Src protein. Sticks represent the three-dimensional structure of daidzein; the surface represents the protein structure of Src protein.

Src protein may be the target of daidzein

In order to explore whether daidzein can effectively combine with Src and play a regulatory role, we assessed the potential interaction between daidzein and Src through molecular modeling and docking simulation. The 3D structure of daidzein was obtained from the PubChem database (Fig. 5E). After adjustment and analysis, the grid size was set to 40 × 40 × 40, and we obtained an affinity of −9.1 kcal/mol. As shown in Fig. 5F, hydrophobic interactions were formed between daidzein and the following amino acid residues: Ile336, Thr338, Met314, Val323, Phe405, Ala403, Leu393, Asp404, Val281 and Ala293. In brief, these results suggest a strong potential of direct binding between daidzein and Src. Thereafter, we employed GO annotation for the functions of the key target genes. The top 20 highly enriched functional terms are shown in Fig. 6A. The key target genes could regulate multiple biological processes, including steroid hormone receptor activity, nuclear receptor activity, ligand-activated transcription factor activity, etc. The KEGG results indicated that these key target genes were involved in multiple signaling pathways, and the top 20 pathways are shown in Fig. 6B. Among them, the MAPK signaling pathway was the highest-ranked pathway, with the highest gene ratio as well as the third minimum adjusted P-value (less than 1e−07). According to previous studies, the MAPK (JNK/ERK/p38) signaling pathway is a downstream molecule of Src kinase (Cui et al., 2017; Gao et al., 2019). Therefore, we further predicted that the Src and MAPK pathways may be involved in the regulation of daidzein on biological process of osteosarcoma.

Figure 6 Daidzein inhibits phosphorylation of Src and ERK in osteosarcoma cells.

(A) The top 20 highly enriched functional terms analyzed by GO enrichment. (B) The top 20 highly enriched signaling pathways analyzed by KEGG enrichment. (C) Western blot results of the Src-MAPK pathway in 143B and U2OS cells treated with different concentrations of daidzein. (D) Quantitative analysis of western blot. Solid bars and errors represent the mean ± SEM, Individual values were shown as dots of different shapes. Student’s t-test, ∗p < 0.05, ∗∗p < 0.01 versus control group.

Daidzein inhibits phosphorylation of the Src and ERK signaling pathways osteosarcoma cells

According to the results of previous systemic pharmacology analysis and molecular docking studies, we detected the protein expression and phosphorylation levels of Src, JNK, ERK and p38 to elucidate the molecular mechanisms by which daidzein affects osteosarcoma cells. As shown in Figs. 6C–6D, daidzein treatment decreased the expression of p-Src and p-ERK, but the expression levels of Src, JNK, p-JNK, ERK, p38 and p-p38 were not significantly changed. Therefore, we determined that Src and ERK phosphorylation was suppressed in daidzein-treated cells, whereas JNK and p38 phosphorylation was not altered.

Overexpression of Src rescues inhibitory effect of daidzein on osteosarcoma cells

As shown in Figs. 7B–7C, overexpression of Src partially restituted the phosphorylation of Src and ERK in osteosarcoma cells treated with daidzein. Moreover, according to MTT assay (Fig. 7D) and apoptosis analysis by flow cytometry (Figs. 7E–7F), decreased viability and in osteosarcoma cells treated with daidzein was partially reversed by overexpression of Src. Therefore, it suggests that daidzein may exert anti-osteosarcoma efficiency through the inhibition of the Src kinase-mediated ERK pathway.

Figure 7 Daidzein exerts antitumor effects through the Src-ERK pathway in osteosarcoma confirmed by the Src overexpression rescue experiments.

(A) The expression of the green fluorescent protein EGFP in Src-overexpressing 143B and U2OS cells was observed under a fluorescence microscope. (B) Western blot results of the Src-ERK pathway in the Src overexpression rescue experiments. (C) Quantitative analysis of western blot of the Src overexpression rescue experiments. (D) MTT analysis of the Src overexpression rescue experiments. (E) Apoptotic cells were detected by the V-FITC/PI double staining assay in the Src overexpression rescue experiments. (F) Quantitative analysis of apoptotic cells from three independent experiments. (Solid bars and errors represent the mean ± SEM, Individual values were shown as dots of different shapes. Student’s t-test, ∗p < 0.05, ∗∗p < 0.01, ∗∗∗p < 0.001 versus control group).

Discussion

Daidzein is often found in beans such as soybeans, sweet red beans, and kidney beans. Previous studies have shown that it has antioxidative, anti-inflammatory, and anti-aging activities (Ahmad et al., 2016; Choi et al., 2012; Liu et al., 2009; Tomar et al., 2020; Yi et al., 2019; Yu et al., 2020; Živanović et al., 2019). Additionally, daidzein has garnered interest as a natural compound with antitumor effects against a variety of tumors. Several previous studies have investigated the antitumor effects and mechanisms of daidzein. For example, in colon cancer, daidzein induces apoptosis of tumor cells by inhibiting the accumulation of lipid droplets (Liang et al., 2018). In bladder cancer, daidzein exerts antitumor activity via inhibition of the FGFR3 pathway (He et al., 2016). Additionally, daidzein has also been shown to enhance the anticancer effect of topotecan and reverse BCRP-mediated drug resistance in breast cancer (Guo et al., 2019), and it inhibits lung metastasis in mice induced by B16F-10 melanoma cells (Menon et al., 1998). In malignant glioma, treatment with subtoxic doses of daidzein in combination with TRAIL induces rapid apoptosis in glioma cells because daidzein overcomes TRAIL resistance by modulating the expression of the intrinsic apoptotic inhibitor bcl-2 (Siegelin et al., 2009).

As a kind of nonreceptor protein tyrosine kinase, Src is an important oncogenic biological factor in a variety of malignancies, and the active form of Src is phosphorylated Src (p-Src) (Dehm & Bonham, 2004). Src interacts with several protein-tyrosine kinase receptors at the plasma membrane, including EGFR, erbB2, c-Met, PDGFR, IGFR and FGFR. These protein-tyrosine kinase receptors are dysregulated or mutated in a variety of human malignancies, including and not limited to NSCLC, breast cancer, pancreatic cancer, colorectal cancer, bladder cancer, prostate cancer, kidney cancer, ovarian cancer and sarcoma and so on Roskoski Jr (2015). Src and that of its protein-tyrosine kinase receptors interact with each other. In addition to protein tyrosine kinases, Src-family kinases are also regulated by integrin receptors, G-protein coupled receptors, antigen- and Fc-coupled receptors, cytokine receptors, and steroid hormone receptors (Thomas & Brugge, 1997). As the central hub of multiple signaling pathways, Src is involved in the amplification cascade of the intracellular network regulation system, whose downstream pathways include FAK, Akt, Stat3 and MAPK. Src is also involved in the regulation of bone formation and remodeling and may mediate skeletal metastasis of breast, prostate, and lung cancers (Roskoski Jr, 2015). Our previous clinical studies have shown that Src/p-Src are potential biomarkers of osteosarcoma, which could promote the progression or metastasis of osteosarcoma and may lead to a poorer prognosis in clinical patients (Hu et al., 2015).

In modern civilization, the spectrum of human diseases has changed greatly, from infectious diseases to complex diseases such as tumors, cardiovascular diseases, and diabetes. The occurrence and development of complex diseases often involve multiple genes and signaling pathways in the regulatory network of the organism. Systemic pharmacology aims at biological networks and analyzes the relationship among the drugs, targets and diseases in these networks (Yuan et al., 2017), which is consistent with the treatment concept of complex diseases: to carry out new drug research and drug mechanistic research. The large amount of experimental material and reagents used in previous screening studies has been saved due to the availability of data from the Internet for analysis, thereby greatly saving time and expense.

Conclusions

In conclusion, through a combination of systemic pharmacology, molecular docking and experimental validation, this study provides powerful proof that daidzein has little negative effect on normal cells and exerts an antitumor effect through the Src-ERK axis, which suggests that daidzein could act as a potential osteosarcoma chemotherapy agent. However, this study has its limitations. First, owing to limitations of time and funds, instead of experimentally validating all target molecules and pathways screened by the network pharmacological analysis, we chose the one with the highest match possibility. Second, we only examined the anti-osteosarcoma effects of daidzein, but we did not validate the mechanism in vivo. In addition, depending on the different cell types and stimuli, MAPKs have a dual role since they can act as activators or inhibitors of apoptosis (Yue & López, 2020). Therefore, further indepth study is needed on the specific mechanism by which daidzein inhibits ERK phosphorylation and thus causes apoptosis in osteosarcoma cells. In addition, more basic and clinical studies are needed to establish the role of daidzein in the treatment of osteosarcoma.

Supplemental Information

Supplemental Information 1 243 human genes associated with daidzein were predicted by the PharmMapper platform

Click here for additional data file.

Supplemental Information 2 4155 osteosarcoma-related genes were collected and combined from DigSeE, DisGeNET, GeneCards, KEGG disease, MalaCards, and Phenopedia

Click here for additional data file.

Supplemental Information 3 133 common targets of osteosarcoma and daidzein

Click here for additional data file.

Supplemental Information 4 Full (21-point) ARRIVE 2.0 checklist

Click here for additional data file.

The authors thank Dr. Xiaobin Zhu and Bin Ren from the Department of Spine Surgery & Musculoskeletal Tumor, Zhongnan Hospital of Wuhan University, China for assisting in experimental design.

Additional Information and Declarations

Competing Interests

Author Contributions

Animal Ethics

Data Availability

The authors declare there are no competing interests.

Yufan Zhu conceived and designed the experiments, performed the experiments, analyzed the data, prepared figures and/or tables, and approved the final draft.

Zhiqiang Yang performed the experiments, analyzed the data, prepared figures and/or tables, and approved the final draft.

Yuanlong Xie conceived and designed the experiments, prepared figures and/or tables, and approved the final draft.

Min Yang performed the experiments, prepared figures and/or tables, and approved the final draft.

Yufeng Zhang and Zhouming Deng conceived and designed the experiments, authored or reviewed drafts of the paper, and approved the final draft.

Lin Cai conceived and designed the experiments, authored or reviewed drafts of the paper, financial support, and approved the final draft.

The following information was supplied relating to ethical approvals (i.e., approving body and any reference numbers):

The study was conducted according to the guidelines of the Declaration of Helsinki, and approved by the Institutional Review Board (or Ethics Committee) of Institutional Animal Care and Use Committee, Wuhan University Center for Animal Experiment (protocol code WP2020-08106. On August 10th, 2020).

The following information was supplied regarding data availability:

The full-length images of western blots are available in the Supplementary File.

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
