# Peer review of "Investigation of inhibition effect of daidzein on osteosarcoma cells based on experimental validation and systematic pharmacology analysis"

_PeerJ, doi:10.7717/peerj.12072_

## Round 0.1 · original submission · Major Revisions

The manuscript is quite interesting and well designed, as pointed by both reviewers. Nonetheless, there are some points raised by the reviewers that need to be addressed before it can be considered suitable for publication in PeerJ.

Reviewer 1 ·

Basic reporting

Zhu and colleagues reported an interesting study on the cellular and molecular effects of treatment with daidzen, an isoflavonoid, in osteosarcoma models.

Experimental design

The experiments were conducted properly.

Validity of the findings

The results support the conclusion of the study.

Additional comments

Zhu and colleagues reported an interesting study on the cellular and molecular effects of treatment with daidzen, an isoflavonoid, in osteosarcoma models. The experiments were conducted properly and the results support the conclusion of the study. I highlight the findings of the systematic pharmacology analysis that allowed the identification of a potential mechanism of action that partially explains the observed results. The figures are adequate and the text is clear and well written. This reviewer suggests minor corrections to avoid misinterpretation by the readers.

1-) In the abstract (Line 19), the authors must include a sentence presenting the daidzen. Make it clear to readers to which class this compound belongs.

2-) In item 6 of the "Material & Methods" the "Tumor invasion assay" assay is actually a "Transwell migration assay", as there is no addition of extracellular matrix (e.g. matrigel) for the cells to invade.

3-) In item 7 of the “Material & Methods” the “Tumor migration assay”, the most appropriate term (to avoid conflict with item 6) would be “Wound assay”.

4-) After correcting items 6 and 7 of the “Material & Methods”, the terms migration and invasion need to be revised throughout the text (e.g. line 252).

5-) Describe the catalog number and the phosphorylation sites evaluated for each antibody described in item 11 of the “Material & Methods”.

6-) The authors use the units of the drug in "uM" and "umol/L". Although the terms are synonymous, the authors must be consistent throughout the text to avoid misinterpretation by the readers.

7-) Authors must include more details in all Figure legends. Readers need to be able to interpret the figures without having to read the entire text. For example, in Figure 1, include the description of the rationale for the study in the legend, the website links and the indication of the software. In all Figures, describe the statistical method and indicate p-values. In Figure 5D, what does the scale bar of the graph mean? Please carefully review all figure legends.

Reviewer 2 ·

Basic reporting

No comment. Please see below

Experimental design

No comment. Please see below

Validity of the findings

No comment. Please see below

Additional comments

The manuscript entitled “Investigation of inhibition effect of daidzein on osteosarcoma cells based on experimental validation and systematic pharmacology analysis” by Zhu explored the effect of daidzein on osteosarcoma and the potential molecular mechanism. The paper has been well-designed and evaluated; but some points should be improved before publication:

1. Some grammatical/formatting errors must to be corrected in the entire text.
2. Regarding the quality control of the cell line, the presence of mycoplasma was investigated in cell culture? Was cell culture negative for mycoplasma?
3. The IC50 values of daidzein on 143B and U2OS cells are 63.59 μmol/L and 125 μmol/L. This compound showed low potency. What is the clinical importance of this? A positive control should be include.
4. Normal cell line (or primary cell culture) should be used to assess the selectively of daidzein.
5. In vitro invasion and migration assays must be carried out using non-cytotoxic concentrations. The interpretation of these assays using ½IC50, IC50 and 2xIC50 is not appropriated.
6. In figure 1, use “IN VIVO experiments” instead “IN VITRO experiments” for xenograft model.
7. For all photograph from microscopy, include more details, e.g. bar scale, total amplification, etc. Add more details in all figure’s legend.
8. Add statistical analysis in figures 2, 3 and 6. Add statistical at the legend of figure 4.
9. The quantification of WB figures is stimulated. Include the MW in each protein of WB.
10. In the experiments with overexpression of Src, the reversion is not clear. The negative control also presented more cells that are viable. I suggest you using a more sensitive assay than MTT (like annexin-V/PI staining by flow cytometry).

---

## Round 0.2 · accepted · Accept

The authors revised their manuscript, answering all the concerns raised by the referrers. The manuscript is well designed and brings novelty to the research field. Thus I agree that the revised manuscript should be published in PeerJ.

Reviewer 1 ·

Basic reporting

The current version of the manuscript has been improved in grammar compared to the previous version. Some errors in the description of the methods were also remedied. The literature review, the article structure and the presentation of results are adequate.

Experimental design

The experimental design is also adequate. The methodological detailing allows the replication of experiments by other researchers.

Validity of the findings

The results presented are interesting and allow for advances in the field of study in which the study is inserted.

Additional comments

In general, the corrections made by the authors substantially improved the current version of the manuscript.

Reviewer 2 ·

Basic reporting

The authors have answered all the questions I concerned, and provided abundant data to support their conclusions.Thus, the manuscript can be accepted for publication.

Experimental design

Above

Validity of the findings

Above

Additional comments

Above